# Propofol Requirement in Patients with Growth Hormone-Secreting Pituitary Tumors Undergoing Transsphenoidal Surgery

**DOI:** 10.3390/jcm8050571

**Published:** 2019-04-26

**Authors:** Seung Hyun Kim, Namo Kim, Eui Hyun Kim, Sungmin Suh, Seung Ho Choi

**Affiliations:** 1Department of Anesthesiology and Pain Medicine, Severance Hospital, Yonsei University College of Medicine, Seoul 03722, Korea; anesshkim@yuhs.ac (S.H.K.); SUH5701@yuhs.ac (S.S.); 2Department of Anesthesiology and Pain Medicine, Anesthesia and Pain Research Institute, Severance Hospital, Yonsei University College of Medicine, Seoul 03722, Korea; NAMO@yuhs.ac; 3Department of Neurosurgery, Pituitary Tumor Center, Severance Hospital, Yonsei University College of Medicine, Seoul 03722, Korea; EUIHYUNKIM@yuhs.ac

**Keywords:** propofol, general anesthesia, transsphenoidal surgery, growth hormone-secreting pituitary tumor

## Abstract

Growth hormone (GH) secretion is regulated by various hormones or neurotransmitters, including gamma-aminobutyric acid. The aim of this study was to determine the propofol requirement in patients with GH-secreting pituitary tumors undergoing transsphenoidal surgery. General anesthesia was induced in 60 patients with GH-secreting tumors (GH group, *n* = 30) or nonfunctioning pituitary tumors (NF group, *n* = 30) using an effect-site target-controlled intravenous propofol infusion. The effect-site concentrations were recorded at both a loss of consciousness and a bispectral index (BIS) of 40, along with the effect-site concentration after extubation, during emergence from the anesthesia. The effect-site concentration of propofol was higher in the GH group than in the NF group at a loss of consciousness and a BIS of 40 (4.09 ± 0.81 vs. 3.58 ± 0.67, *p* = 0.009 and 6.23 ± 1.29 vs. 5.50 ± 1.13, *p* = 0.025, respectively) and immediately after extubation (1.60 ± 0.27 vs. 1.40 ± 0.41, *p* = 0.046). The total doses of propofol and remifentanil during anesthesia were comparable between the groups (127.56 ± 29.25 vs. 108.64 ± 43.16 µg/kg/min, *p* = 0.052 and 6.67 ± 2.89 vs. 7.05 ± 1.96 µg/kg/h, *p* = 0.550, respectively). The propofol requirement for the induction of a loss of consciousness and the achievement of a BIS of 40 is increased during the induction of general anesthesia in patients with GH-secreting tumors.

## 1. Introduction

Growth hormone-secreting pituitary tumors are the most common cause of acromegaly, which is associated with serious complications, including cardiomyopathy and heart failure. Transsphenoidal pituitary surgery is the standard surgical treatment for most pituitary tumors [1]. A smooth emergence from general anesthesia is important after this type of surgery to limit the risk of complications, such as a cerebrospinal fluid leak [2]. Propofol is known to be better than inhalation agents for reducing the incidence of agitation and delirium after general anesthesia and is widely used in transsphenoidal pituitary surgery. Propofol exerts a sedative effect by activating the receptor for gamma-aminobutyric acid (GABA), a major inhibitory neurotransmitter in the central nervous system [3,4,5]. The pharmacokinetics of propofol can be altered by various factors, including age, sex, weight, concomitant medications, and underlying diseases, so it is important to tailor the dose to the individual patient’s requirements [6].

It is well known that the secretion of growth hormones from the pituitary gland is regulated primarily by hormones released from the hypothalamus, i.e., growth hormone-releasing hormone and growth hormone-inhibiting hormone. The hypothalamic pituitary neuroendocrine axis is controlled by complex networks of various hormones, and hypothalamic GABA can stimulate anterior pituitary hormone secretion, including that of growth hormones [7,8,9,10]. Furthermore, the synthesis of GABA within the pituitary gland has been reported, which suggests that pituitary GABA has an autocrine function [11]. 

We speculated that the interaction between GABA and growth hormones could affect the action of propofol in patients with acromegaly. The aim of this study was to determine if there is a difference in the propofol requirement between patients with growth hormone-secreting pituitary tumors and those with nonfunctioning pituitary tumors.

## 2. Materials and Methods

The study protocol was approved by the Institutional Review Board of the Yonsei University Health System, Seoul, South Korea (#4-2017-1241) and is registered at ClinicalTrials.gov (NCT03465423). Written informed consent was obtained from all patients enrolled in the study, which was conducted from February 2018 to November 2018. The study population consisted of 60 patients aged over 19 years who underwent elective transsphenoidal pituitary surgery for a nonfunctioning or growth hormone-secreting tumor. Patients with an abnormal pituitary function, including hypopituitarism, prolactinoma, Cushing disease, and hypothyroidism were excluded. Patients who had significant heart disease (e.g., heart failure, arrhythmia, and myocardial infarction), renal disease, suffered a recent stroke, a psychiatric illness, a cognitive disorder, a propofol allergy, or who were pregnant were also excluded. 

On arrival in the operating room, a bispectral index (BIS) monitor (BIS VISTA Monitoring System, Aspect Medical Systems Inc., Norwood, MA, USA) was applied to assess the depth of the anesthesia. Standard intraoperative monitoring, including pulse oximetry, noninvasive blood pressure monitoring, and electrocardiography, was performed in all cases. All patients were premedicated with 0.2 mg glycopyrrolate (Glycopyrrolate; Reyon pharm Co., Seoul, Korea). Total intravenous general anesthesia was induced using an effect-site target-controlled infusion pump (Orchestra Base Primea; Fresenius Vial, Brézins, France). Propofol (Fresofol MCT 2%; Fresenius Kabi, Seoul, Korea) was administered according to the Marsh model and remifentanil (Ultian; Hanlim pharm Co., Seoul, Korea) according to the Minto model [12,13,14]. The initial target effect-site concentrations (Ces) for propofol and remifentanil were 3 µg/mL and 3 ng/mL, respectively. The level of consciousness was assessed using a modification of the Observer Assessment of Awareness and Sedation Scale (OAA/S). When the patient did not respond to loud and repeated verbal commands (OAA/S score < 2, i.e., indicating a loss of consciousness (LOC)), rocuronium 0.8 mg/kg was administered. We steadily increased the target Ce of propofol by increments of 0.5 µg/mL, while maintaining the Ce of remifentanil at 3 ng/mL, until the BIS value reached 40. The Ce of propofol was recorded at both a LOC and a BIS of 40. The durations from the start of the propofol administration to LOC and BIS of 40 were also recorded. After tracheal intubation, the Ce of propofol was adjusted to maintain a BIS value in the range of 40–60. Arterial blood pressure was monitored via a radial artery catheter and mechanical ventilation was maintained with 50% oxygen. End-tidal CO_2_ and esophageal temperature were maintained in the ranges of 35–40 mmHg and 36–37 °C, respectively. 

The BIS score was recorded during the emergence phase before the discontinuation of propofol and remifentanil. The Ce of propofol was recorded when the patients opened their eyes to verbal command and immediately after extubation. The interval between the discontinuation of propofol and extubation was also recorded. 

### Statistical Analysis

The standard deviation of the Ce of propofol at a LOC in patients undergoing neurosurgery has been reported to be 0.35 µg/mL [15]. Assuming a between-group difference > 0.3 µg/mL to be statistically significant, we calculated that 62 patients would need to be enrolled for a power of 0.8. 

The descriptive data are presented as the mean ± standard deviation. For between-group comparisons, either the chi-square test or the Fisher’s exact test was used for categorical variables and either the Student’s *t*-test or the Wilcoxon–Mann–Whitney test was used for continuous variables. All statistical analyses were performed using SPSS Statistics for Windows (version 23; IBM Corp., Armonk, NY, USA).

## 3. Results

Two of the 62 patients deemed eligible for this study declined to participate. Therefore, 60 patients with growth hormone-secreting tumors (GH group, *n* = 30) or nonfunctioning tumors (NF group, *n* = 30) were finally enrolled. The patient height, weight, and body mass index were higher in the GH group than in the NF group because of the characteristics of these tumors. Otherwise, there were no significant between-group differences in the clinical data, including for the duration of anesthesia, tumor size, sinus invasion of the tumor, intraoperative fluid requirement, bleeding, and urine output (Table 1). 

During the induction of anesthesia, the Ce of propofol at the time of LOC was higher in the GH group than in the NF group (4.09 ± 0.81 vs. 3.58 ± 0.67; *p* = 0.009). The Ce of propofol at a BIS of 40 was also higher in the GH group than in the NF group (6.23 ± 1.29 vs. 5.50 ± 1.13; *p* = 0.025; Table 2). The intervals between the start of the administration of propofol to LOC and BIS of 40 were comparable between the two groups (2.25 ± 0.94 min vs. 1.83 ± 0.82 min; *p* = 0.070, 5.42 ± 2.30 min vs. 4.65 ± 1.72 min; *p* = 0.151). 

During emergence from the anesthesia, the Ce of propofol was higher in the GH group than in the NF group immediately after extubation (1.60 ± 0.27 vs. 1.40 ± 0.41, *p* = 0.046). The BIS value at the time of extubation was lower in the GH group than in the NF group (70.41 ± 19.76 vs. 79.53 ± 7.92, *p* = 0.026). However, the mean Ce of propofol when the patients were able to respond to verbal commands was comparable between the two groups (1.62 ± 0.28 vs. 1.44 ± 0.44, *p* = 0.089). 

The total dose of propofol during anesthesia was comparable between the two groups (127.56 ± 29.25 µg/kg/min vs. 108.64 ± 43.16 µg/kg/min; *p* = 0.052). The total dose of remifentanil was also comparable between the two groups (6.67 ± 2.89 µg/kg/h vs. 7.05 ± 1.96 µg/kg/h; *p* = 0.550).

## 4. Discussion

This study has demonstrated that the propofol requirement for a LOC and the achievement of a BIS of 40 is higher in patients undergoing transsphenoidal surgery for a growth hormone-secreting pituitary tumor than in their counterparts with a nonfunctioning pituitary tumor. The Ce of propofol at the time of a LOC and a BIS of 40 was significantly higher in the GH group than in the NF group. 

These findings are clinically significant, because they show that a higher dose of propofol is required to achieve an adequate depth of anesthesia during the induction and maintenance of the total intravenous general anesthesia in these patients. It is also noteworthy that extubation was performed at a higher concentration of propofol and at a higher BIS value in the GH group than in the NF group. A smooth emergence from anesthesia is essential in patients undergoing transsphenoidal surgery to avoid complications, such as cerebrospinal fluid leak [2]. Accordingly, the recognition of the high propofol requirement in these patients would be important for patient safety. Previous studies of the management of general anesthesia in patients with acromegaly have focused mainly on airway difficulties and coexisting disease, including cardiac complications [16,17,18]. To our knowledge, there have been no reports on the optimal propofol dose for an adequate depth of anesthesia, and this is the first study to demonstrate an increased propofol requirement in this population. 

The mechanism of the action of propofol is not fully understood but is well known to be related to the potentiation of the gamma-aminobutyric acid A (GABA_A_) receptor [3,4,19]. The GABA_A_ receptor is a pentamer composed mostly of three subunits (α, β, and γ). Although there are numerous isoforms of the GABA_A_ receptor as a result of the various combinations of the subunits, the β subunit plays a major role in the action of propofol [3,20]. Propofol binds to the GABA receptor and reduces neuronal excitability by increasing the Cl^−^ current induced by GABA. In high concentrations, propofol can also directly activate the GABAA receptor [21]. GABA stimulates the secretion of growth hormones via pituitary GABA receptors and is one of the most important inhibitory neurotransmitters in the central nervous system [11]. The activity of GABA in patients with acromegaly may be different from that in the normal population and may interfere with the action of propofol. However, the mechanism by which the propofol requirement in patients with growth hormone-secreting tumors is increased is unclear.

Pituitary GABA is known to be synthesized in the hypothalamus and secreted into the pituitary gland via the hypophyseal portal vessels [22]. GABA_A_ receptors are distributed in the endocrine cells in the anterior of the pituitary gland. The synthesis and secretion of GABA also occur in the pituitary gland itself, implying that GABA acts in an autocrine or paracrine fashion in this gland [11]. When GABA or its agonists bind to the GABA_A_ receptor simultaneously, their effects may be altered by their interaction, which relates to the location of the subunits to which each agonist binds [23]. The effect of propofol is also affected by the locations of the GABA_A_ receptor on the cell surface [21]. GABA_A_ receptors are located mainly on the postsynaptic membrane; however, extrajunctional GABA_A_ receptors are also suggested to be important targets for sedative agents [24]. The extrajunctional GABA_A_ receptors, in particular, have been implicated in the action of propofol [25]. The response of the GABA_A_ receptors to sedative agents may be altered when they are activated. According to one study, the action of propofol on extrajunctional GABA_A_ receptors is only effective when the ambient concentration of GABA is low [26]. The extrajunctional GABA_A_ receptor is likely to be more sensitive to sedative agents in conditions of low GABA occupancy [24]. Considering the role of GABA as an important stimulator of growth hormone secretion, the concentration of ambient GABA in patients with acromegaly may be higher than that in the normal population, and this could be one of the reasons for the increased propofol requirement in this population. 

This study has some limitations. First, we did not measure the plasma concentrations of propofol in our patients and used the Marsh model to determine its effect-site concentration. However, the pharmacokinetics in patients with a growth hormone-secreting tumor may still be different from those in patients with a nonfunctioning tumor and those in the normal population. To our knowledge, there have been no studies on the pharmacokinetics of propofol in patients with acromegaly, and further investigations are needed. Second, the height, weight, and body mass index of the patients were not comparable between the two groups due to the characteristics of these tumors. However, propofol was administered according to the Marsh model, in which the dose was adjusted based on height and weight. Thirdly, the study was underpowered in terms of the total dose of propofol administered during anesthesia, which was one of the secondary outcomes in our study. The actual power of the total propofol dose in this study was 0.78. Therefore, the total propofol dose during the anesthesia in the GH group was higher than that in the NF group; however, the difference was not statistically significant. 

## 5. Conclusions

The propofol requirement for a LOC and the achievement of a BIS of 40 are increased during the induction of general anesthesia in patients with a growth hormone-secreting tumor, meaning that a higher dose of propofol is needed to achieve an adequate depth of anesthesia in this population. Further studies elucidating the underlying mechanism are warranted.

## Figures and Tables

**Table 1 jcm-08-00571-t001:** Patient demographics and clinical data in this study.

	Nonfunctioning Tumor (*n* = 30)	GH-Secreting Tumor (*n* = 30)	*p*-Value
Age (year)	50.47 ± 11.79	44.53 ± 12.15	0.060
Sex (male/female)	19/11	22/8	0.405
Height (cm)	164.86 ± 9.58	170.10 ± 11.63	0.027 *
Weight (kg)	67.42 ± 10.92	78.04 ± 16.42	0.005 *
BMI	24.69 ± 2.45	26.40 ± 3.19	0.023 *
ASA (1/2/3)	15/12/3	0/24/6	<0.001 *
Duration of surgery (min)	168.65 ± 58.29	170.58 ± 70.58	0.919
Duration of anesthesia (min)	261.03 ± 65.99	247.83 ± 76.34	0.477
Tumor size (>1 cm)	29 (96.7%)	24 (80%)	0.103
Sinus invasion	11 (36.7%)	7 (23.3%)	0.260
Fluid intake (mL)	1525.00 ± 601.40	1735.67 ± 665.94	0.204
Bleeding (mL)	225.67 ± 147.41	321.67 ± 288.18	0.110
Urine output (mL)	443.67 ± 257.81	438.33 ± 344.93	0.946
Total propofol dose (µg/kg/min)	108.64 ± 43.16	127.56 ± 29.25	0.052
Total remifentanil dose (µg/kg/h)	7.05 ± 1.96	6.67 ± 2.89	0.550

Data are presented as mean ± SD for continuous variables and count (percentage) for categorical variables. GH: growth hormone. * *p* < 0.05 compared with nonfunctioning tumor group. ASA, American Society of Anesthesiologists physical status classification; BMI, body mass index; SD, standard deviation.

**Table 2 jcm-08-00571-t002:** Comparison of propofol effect-site concentrations between the nonfunctioning tumor patients and the growth hormone-secreting tumor patients.

		Nonfunctioning Tumor (*n* = 30)	GH-Secreting Tumor (*n* = 30)	*p*-Value
at LOC	Ce (µg/mL)	3.58 ± 0.67	4.09 ± 0.81	0.009 *
	Time from start of propofol to LOC (min)	1.83 ± 0.82	2.25 ± 0.94	0.070
	BIS	76.57 ± 11.04	75.50 ± 9.18	0.686
at BIS 40	Ce (µg/mL)	5.50 ± 1.13	6.23 ± 1.29	0.025 *
	Time from start of propofol to BIS 40 (min)	4.65 ± 1.72	5.42 ± 2.30	0.151
Immediately after the extubation	Ce (µg/mL)	1.40 ± 0.41	1.60 ± 0.27	0.046 *
	Time from discontinuation of propofol to extubation (min)	15.63 ± 4.18	14.69 ± 3.99	0.380
	BIS	79.53 ± 7.92	70.41 ± 19.76	0.026 *

Data are presented as mean ± SD. BIS: bispectral index; Ce: effect-site concentration; GH: growth hormone; LOC: loss of consciousness. * *p* < 0.05 compared with nonfunctioning tumor group.

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
