# Peer review of "Propofol Requirement in Patients with Growth Hormone-Secreting Pituitary Tumors Undergoing Transsphenoidal Surgery"

_jcm, 2019, doi:10.3390/jcm8050571_

Round 1
Reviewer 1 Report
This is original and interesting paper regarding general anesthesia induced with propofol in patients with acromegaly. Several improvements could be made as follows: 1. In the Method section size of tumors and/or pituitary function should be specified since functional and non functional macroadenomas can cause hypopituitarism which could also affect parameters of interest. 2. Adjustment analysis should be performed for other factors which could influence propofol concentrations like age, weight, BMI. 3. In a Discussion section it should be hypothesized in what way is GABA concentrations changed (decreased, increased) in acromegaly in authors' opinion thus influencing propofol concentration. Otherwise paper is written clearly, The results, the discussion and conclusion sections are in general adequately and appropriately presented. The tables are easily readable. .
Author Response
We uploaded response to reviewer comments.
This is original and interesting paper regarding general anesthesia induced with propofol in patients with acromegaly. Several improvements could be made as follows:
Thank you very much for your constructive review.
Point 1: In the Method section size of tumors and/or pituitary function should be specified since functional and non functional macroadenomas can cause hypopituitarism which could also affect parameters of interest.
Response 1: Thank you for your valuable comment. As you mentioned, macroadnoma can cause hypopituitarism, and indeed, we excluded the patients with abnormal pituitary function other than growth hormone excess. According to your comment, we clarified that we excluded the patients with abnormal pituitary function including hypopituitarism, prolactinoma, Cushing disease, and hypothyroidism. The following is the added text in the Materials and Methods.
----------------------------------------------------------------------------------------------------------------
The patients with abnormal pituitary function including hypopituitarism, prolactinoma, Cushing disease, and hypothyroidism were excluded.
Point 2: Adjustment analysis should be performed for other factors which could influence propofol concentrations like age, weight, BMI.
Response 2: Thank you for your thorough review, and we apologize for the lack of detailed explanation. In our study, propofol was administered via effect-site target controlled infusion pump according to Marsh model, in which the dose of propofol is adjusted based on age, height, and weight. As for your suggestion, we added detailed explanation about Marsh model in the limitation. The following is the added text.
----------------------------------------------------------------------------------------------------------------
The height, weight, and body mass index of the patients were not comparable between the two groups due to the characteristics of these tumors. However, propofol was administered according to Marsh model, in which the dose of it was adjusted based on height, and weight.
Point 3: In a Discussion section it should be hypothesized in what way is GABA concentrations changed (decreased, increased) in acromegaly in authors' opinion thus influencing propofol concentration. Otherwise paper is written clearly, The results, the discussion and conclusion sections are in general adequately and appropriately presented. The tables are easily readable.
Response 3: We appreciate the kind and thorough review. Considering that GABA can stimulate the secretion of growth hormone, we think, it is plausible that the concentration of ambient GABA in patients with acromegaly might be high. We changed the text according to your comment. The following is the modified text.
----------------------------------------------------------------------------------------------------------------
Considering the role of GABA as an important stimulator of growth hormone secretion, the concentration of ambient GABA in patients with acromegaly could be higher than that in the normal population, and this could be the one of the reasons for the increased propofol requirement in this population.
Reviewer 2 Report
The manuscript presents an interesting study regarding the propofol requirements in patients with growth hormone-secreting tumors compared with those with non-functioning pituitary tumors. The manuscript is well written, the methodology is well described and provide enough details in order the study to be reproduced. The results are clearly presented and the discussions ad conclusions are based on the results obtained. I have the following suggestions for the authors in order to improve the manuscript: In the introduction please check also the following reference: -Impact of Anesthetics on Cardioprotection Induced by Pharmacological Preconditioning. J Clin Med. 2019;8(3). -Optimizing intraoperative administration of propofol, remifentanil, and fentanyl through pharmacokinetic and pharmacodynamic simulations to increase the postoperative duration of analgesia. J Clin Monit Comput. 2019. In the material and method part i suggest to include the manusfacturer also for the drugs administered, eg: glycopyrrolate, propofol, remifentanil. - You specify in the material and methods that the administration of propofol was made according to Marsh model and remifentanil according to the Minto model. You should add references for the 2 models. - In the tables will be more easy if you notify by * where the difference is statistical significant between the groups.
Author Response
We uploaded the response to the reviewer's comment.
The manuscript presents an interesting study regarding the propofol requirements in patients with growth hormone-secreting tumors compared with those with non-functioning pituitary tumors. The manuscript is well written, the methodology is well described and provide enough details in order the study to be reproduced. The results are clearly presented and the discussions ad conclusions are based on the results obtained. I have the following suggestions for the authors in order to improve the manuscript:
We appreciate your kind and constructive review.
Point 1: In the introduction please check also the following reference: -Impact of Anesthetics on Cardioprotection Induced by Pharmacological Preconditioning. J Clin Med. 2019;8(3). -Optimizing intraoperative administration of propofol, remifentanil, and fentanyl through pharmacokinetic and pharmacodynamic simulations to increase the postoperative duration of analgesia. J Clin Monit Comput. 2019.
Response 1: Thank you. We added the articles you recommended as references in the Introduction.
Point 2: In the material and method part i suggest to include the manusfacturer also for the drugs administered, eg: glycopyrrolate, propofol, remifentanil. - You specify in the material and methods that the administration of propofol was made according to Marsh model and remifentanil according to the Minto model. You should add references for the 2 models. - In the tables will be more easy if you notify by * where the difference is statistical significant between the groups.
Response 2: Thank you very much for your constructive comments. We included the manufacturers for the drugs we administered in our script, and added references for the Marsh, and Minto model. We also added * where the differences is statistically significant according to your suggestion.